# Identifying ITGB2 as a Potential Prognostic Biomarker in Ovarian Cancer

**DOI:** 10.3390/diagnostics13061169

**Published:** 2023-03-18

**Authors:** Chanyuan Li, Ting Deng, Junya Cao, Yun Zhou, Xiaolin Luo, Yanling Feng, He Huang, Jihong Liu

**Affiliations:** 1Cancer Center, The Fifth Affiliated Hospital of Sun Yat-sen University, Zhuhai 519000, China; 2Department of Gynecologic Oncology, State Key Laboratory of Oncology in South China, Collaborative Innovation Center for Cancer Medicine, Sun Yat-sen University, Guangzhou 510060, China

**Keywords:** ITGB2, ovarian cancer, prognostic immunomarker

## Abstract

Epithelial ovarian cancer is by far the most lethal gynecological malignancy. The exploration of promising immunomarkers to predict prognosis in ovarian cancer patients remains challenging. In our research, we carried out an integrated bioinformatic analysis of genome expressions and their immune characteristics in the ovarian cancer microenvironment with validation in different experiments. We filtrated 332 differentially expressed genes with 10 upregulated hub genes from the Gene Expression Omnibus database. These genes were closely related to ovarian tumorigenesis. Subsequently, the survival and immune infiltration analysis demonstrated that the upregulation of five candidate genes, ITGB2, VEGFA, CLDN4, OCLN, and SPP1, were correlated with an unfavorable clinical outcome and increased immune cell infiltration in ovarian cancer. Of these genes, ITGB2 tended to be the gene most correlated with various immune cell infiltrations and had a strong correlation with significant M2 macrophages infiltration (r = 0.707, *p* = 4.71 × 10^−39^), while it had a moderate correlation with CD4+/CD8+ T cells and B cells. This characteristic explains why the high expression of ITGB2 was accompanied by immune activation but did not reverse carcinogenesis. Additionally, we confirmed that ITGB2 was over-expressed in ovarian cancer tissues and was mainly located in cytoplasm, detected by Western blotting and the immunohistochemical method. In summary, ITGB2 may serve as a prognostic immunomarker for ovarian cancer patients.

## 1. Introduction

EOC (Epithelial ovarian cancer) is an aggressive gynecological malignancy with high relapse and mortality rates due to the limited early detection tactics and the absence of effectiveness of existing chemo- and immunotherapies. Thus, it is essential to ascertain alternative diagnostic biomarkers and potential novel targets for ovarian cancer therapy [1,2].

Immuno-oncology has been introduced to the combined treatment of some intractable, advanced malignancies for over a decade, including ovarian cancer patients. The evolution of several immune-based therapies has contributed to effective antitumor responses by regulating the host immune system, including through molecular therapy, cellular therapy in the form of vaccines, CAR-T (Chimeric antigen receptor T-Cell) therapy, and immunomodulator therapy as an immune checkpoint blockade [3,4]. Some clinical trials have incorporated ICIs (immune checkpoint inhibitors) into conventional, platinum-based chemotherapy regimens in serous ovarian cancer patients in combination with cytoreductive surgery, with the aim of reducing the patients’ pain and prolonging their survival time [5,6]. Unfortunately, only a minority of patients could benefit from the immune checkpoint blockade treatments through the application of inhibitors of PD-1 (programmed cell death protein 1), such as Sintilimab and Pembrolizumab, and CTLA-4 (cytotoxic T-lymphocyte associated protein 4): Lpilimumab [7,8,9]. Therefore, novel and effective biomarkers to predict and assess the responses to immunotherapy in serous ovarian cancer are required. Additionally, several studies have reported that an effective antitumor response of immune infiltration would be beneficial to the ICIs’ efficacy in patients, and the prognosis of patients particularly depends on the immune infiltration and the epigenetic mutation load of immune cells in the tumor microenvironment [10,11]. Thus, our study intended to demonstrate the possible association between genome expression and tumor-immune interactions in ovarian cancer.

Here, we performed a comprehensive bioinformatic analysis of ITGB2 (integrin beta-2) and its immune-related characteristics in ovarian tumorigenesis. We first found that several genes, including ITGB2, were highly upregulated in serous ovarian cancer samples. Subsequently, the gene and protein interactions with ITGB2 were identified via Metascape and STRING databases, and the high expression of ITGB2 could significantly affect the clinical outcome for patients with serous ovarian cancer. Finally, the TIMER (tumor immune estimation resource) database was employed to evaluate the interactions between ITGB2 and the immune infiltration in the TME (tumor microenvironment). The findings implied that upregulated ITGB2 is particularly related to the infiltration of M2 macrophages. In summary, this study revealed that ITGB2 might function as a prognostic biomarker in ovarian cancer and might lay the groundwork for the landscape of immuno-therapy strategies in advanced serous ovarian cancer.

## 2. Materials and Methods

### 2.1. Microarray Data

We obtained the serous-ovarian-cancer-related gene expression datasets and matched non-ovarian cancer sample datasets from the database GEO (gene expression omnibus): https://www.ncbi.nlm.nih.gov/geo/ (accessed on 20 September 2022). The expression profiles selected were GSE36668 and GSE66957.

### 2.2. DEGs (Differentially Expressed Genes) Identification

We conducted a DEGs analysis between serous ovarian cancers and non-ovarian tumor tissues through the online tool GEO2R via the limma package in the 3.2.3 version R software from https://www.ncbi.nlm.nih.gov/geo/geo2r/ (accessed on 20 September 2022). Genes in the profiles which satisfied the standards of a |logFC| ≥ 2.0 and an adjusted *p* value < 0.05 would be confirmed as DEGs. Furthermore, the upregulated differential genes we defined as logFC > 2, and the logFC of downregulated genes was <−2. A Venn diagram analysis was also utilized to probe the intersection of the selected DEGs from http://bioinformatics.psb.ugent.be/webtools/Venn/ (accessed on 20 September 2022).

### 2.3. Pathway Enrichment and Functional Analysis of the DEGs

Biological function was visualized through a web-based toolkit, Metascape, GO (Gene ontology) enrichment pathways, and KEGG (Kyoto encyclopedia of genes and genomes) -enriched pathways of the genes. Standards included a *p*-value < 0.01 and minimum > 3. A minimum enrichment factor count of 1.5 would be viewed as statistically significant terms and assembled into clusters on the basis of their similar membership. Then, the immune-associated pathways for ITGB2 would be identified using the Genecards platform: https://www.genecards.org (accessed on 20 September 2022) and the WikiPathways module, and a Reactome pathway analysis was performed.

### 2.4. Hub Genes Identification

The network PPI (protein–protein interaction) was formulated using the toolkit Cytoscape (version 3.7.1) by screening out the nodes with a composite score > 0.4 (STRING: https://cn.string-db.org (accessed on 20 September 2022). The top 10 genes in the midst of PPI were elicited via the cytoHubba plug-in using the degree algorithm, and a degree score > 38 was selected as a hub gene in our study.

### 2.5. Hub Genes Prognosis Analysis

The KM (Kaplan–Meier) plotter online toolkit: http://kmplot.com/analysis/ (accessed on 25 September 2022) was utilized to acquire the survival and corresponding clinical information with the criteria: median; hazards ratio: yes; 95% CI (confidence interval): Yes. The hub genes’ expression and the particular correlation between the differential genes and various classical surface markers of the infiltrating immune cells were verified using the online platform GEPIA (gene expression profiling interactive analysis): http://gepia.cancer-pku.cn/index.html (accessed on 25 September 2022). Results with an adjusted HR (hazard ratio), a 95% CI, and a log-rank *p*-value were needed.

### 2.6. Key Genes Immune Infiltration Analysis

To determine the particular correlation between the prognosis-correlated hub DEGs and a variety of tumor-infiltrating immune cells, we utilized toolkits through the appropriate functional modules: EPIC (estimate the proportion of immune and cancer cells): https://gfellerlab.shinyapps.io/EPIC_1-1/ (accessed on 25 September 2022) and the 2.0 version of the TIMER (tumor immune estimation resource 2.0) database from http://timer.cistrome.org (accessed on 25 September 2022).

### 2.7. Immunofluorescence

PDC3 (patient-derived ovarian cancer cell) was obtained from Tao Zhu, Professor of The Cancer Hospital of the University of Chinese Academy of Sciences. The PDC3 cell we received was cultured in DMEM/F12 (Dulbecco’s modified eagle medium/nutrient mixture12) medium containing 10–15% FBS (Fetal bovine serum) and 1% non-essential amino acid, while the IOSE80 cell (normal ovarian epithelial cell) was cultured in 1640 medium. Both cells were cultured at 37 °C with a 5% CO_2_ in a thermostatic incubator. PDC3 and IOSE80 were then incubated with a ITGB2 primary antibody for 2–4 h at 37 °C, followed by an incubation with secondary antibody Alexa Fluor 594 conjugates at 25 °C for 1–2 h. Subsequently, the nucleus and lysosome were stained with Hoechst 33342 a and lysosome-staining kit, respectively, for 10–15 min at room temperature prior to their observation under a confocal microscope (Nikon, A1 HD25, Tokyo, Japan).

### 2.8. Western Blot

Cell lysates were first obtained by scraping the cultured PDC3 and IOSE80, which were then lysed with a cocktail of protease inhibitors in a RIPA (radioimmunoprecipitation assay) lysis buffer on an ice box (4 °C). Next, the boiled proteins were separated through SDS-PAGE (sodium dodecyl sulphate-polyacrylamide gel electrophoresis) and blocked in PBST (phosphate buffer solution with tween) containing 5% nonfat dried milk. This was followed by incubation with an anti-ITGB2 rabbit monoclonal antibody at 4 °C for 12 h overnight, and a HRP (horseradish peroxidase)-coupled secondary antibody for 2 h. Finally, the target protein bands were detected using a protein imaging system (GE ImageQuant800, Fairfield, CT, USA).

### 2.9. Immunohistochemistry Analysis

The tumor tissues of 8 serous ovarian cancer patients and 8 non-ovarian tumor controls were collected. Paraffin-embedded tissue sections were cut continuously at 4–5 mm from the tissues and then dewaxed. Next, the antigen was retrieved through the application of a citrate buffer. The tissue slides were then incubated with ITGB2 (Absin, Shanghai, China) primary antibodies overnight or for 12 h at 4 °C. After the 2 h incubation with the secondary antibodies, an ABC (Avidin-biotin complex) Substrate System (Servicebio, Wuhan, China) was applied on the slides for the coloration reaction and with HE (hematoxylin) for counterstaining. Images of the slides and staining were eventually acquired on an Olympus VS200 microscope at 20× magnification. All the quantifications were performed with ImageJ, using IHC (immunohistochemistry) profiler plugins. Positive signals in the epithelial tissue were selected for analysis.

### 2.10. QRT-PCR (Quantitative, Real Time-Polymerase Chain Reaction)

The total RNA (ribonucleic acid) of the PDC3 and IOSE80 cells was isolated according to the standard procedure with the reagent kit (Vazyme, Nanjing, China) FastPure cell-Tissue Total RNA Isolation Kit V2. The primer sequence of ITGB2 was F5′ ATGTAAGTGGCCGTCCTTGG 3′, R5′ GGAAGCCGTCACTTTGAGGA 3′. All qPCR experiments were performed by employing the reagent kit (Vazyme, Nanjing, China) HiScript II One Step qRT-PCR SYBR Green Kit and the software Light Cycler96 (Roche, Basel, Switzerland). At least three replicates were used to obtain each average Ct (cycle threshold) value.

### 2.11. Statistics Analysis

We utilized GraphPad Prism 9.0 to conduct a statistical analysis between the serous ovarian cancer group and the matched non-ovarian tumor group. For parametric data, a two-tailed, unpaired Student’s *t* test was utilized to determine the statistical significance between the two groups. With the data not applicable to a normal distribution, we turned to the nonparametric test, and the Mann–Whitney U test was used to analyze the statistical differences between the groups with the exact method.

## 3. Results

### 3.1. DEGs Identification

In this research, we first selected two expression profiles from the GEO database for the subsequent analysis: GSE66957 and GSE36668. The GSE66957 profile consisted of 57 epithelial ovarian cancer tissues and 12 non-tumor tissues, while the GSE36668 encompassed four epithelial ovarian cancer tissues and four matched, non-tumor tissue samples. A volcano plot analysis identified significantly differentially expressed genes in the two datasets, with a standard of *p* value < 0.05 and |logFC| ≥ 2 (Figure 1a,b). In total, 1393 DEGs were screened out from the GSE36668. Among these, 732 genes were upregulated and 661 genes were downregulated significantly in serous ovarian cancer tissues. In the GSE66957 profile, 2578 DEGs were screened out, with 1852 upregulated genes and 726 downregulated genes in serous ovarian cancer tissues. All the DEGs were filtered by comparing the epithelial ovarian cancer tissues to the non-ovarian tumor groups. A Venn analysis was then employed to acquire the intersection of these differential genes (Figure 1c,d). Overall, 332 DEGs, including 301 upregulated genes and 31 downregulated genes, were discovered, and the expression heatmap of the genes showed that the over- or under-expression of these DEGs was highly correlated with serous ovarian cancer (Figure 1e,f).

### 3.2. Analysis of Gene Function and Pathway Enrichment

As we discovered 332 DEGs related to serous ovarian cancer, a series of analyses based on gene annotation and pathway enrichment on the above 332 DEGs were conducted via Metascape. Analyses were conducted for the sake of exploring the potential biological function of the differential genes. Results suggested that these DEGs were majorly enriched in the regulation of cell adhesion, cell–cell adhesion, cell junction organization, the integrin-mediated signaling pathway, and extracellular matrix organization (Figure 2a,b). The details of the annotation and enrichment information of the 332 DEGs were presented in Appendix A. In conclusion, the serous-ovarian-cancer-related DEGs we identified may play a pivotal role in cell communications and transitions.

### 3.3. Identification of Ten Hub Genes through PPI

The hub genes of the DEGs were further selected using STRING and Cytoscape. The DEGs’ interactional proteins were first predicted via the STRING database, and then the densest connection modes were analyzed using the Cytoscape platform. The top ten genes in the PPI network were CDH1 (Cadherin 1), EPCAM (epithelial cell adhesion molecule), ITGB2, CLDN7 (Claudin7), VEGFA (vascular endothelial growth factor A), MUC1 (polymorphic epithelial mucin1), CLDN4 (Claudin4), NANOG (Nanog homeobox), OCLN (Occludin), CDKN2A (Cyclin-dependent kinase inhibitor 2A), LYN (LYN proto-oncogene, Src family tyrosine kinase), and SPP1 (secreted phosphoprotein 1) (Appendix A). The hub genes’ expression differences between the serous ovarian cancer tissues and the non-ovarian tumor tissues in the two profiles a shown in Figure 1e,f). Among these, DH1, VEGFA, EPCAM, ITGB2, and CLDN7 had the highest correlation scores with serous ovarian cancer.

### 3.4. ITGB2, VEGFA, CLDN4, OCLN, and SPP1 Were Correlated with Poor Prognosis in Serous Ovarian Cancer Patients

To seek the candidate biomarkers with prognostic potency, we performed an analysis of the capacity of the ten candidate genes to predict the prognosis of patients via the online Kaplan–Meier plotter platform. Interestingly, we discovered that the PFS (progression-free survival) was significantly reduced in ovarian cancer patients when the expression level of the related genes was high (*p* < 0.05), as were the related hub genes, including ITGB2 (HR = 1.24, *p* = 0.0027), VEGFA (HR = 1.38, *p* = 1.8 × 10^−5^), CLDN4 (HR = 1.22, *p* = 0.0047), OCLN (HR = 1.38, *p* = 9.1 × 10^−6^), and SPP1 (HR = 1.38, *p* = 7.7 ×10^−7^). This indicates that the active transcription of these genes might cause risks in tumorigenesis. Therefore, the five genes might have the potential to be alternative prognostic biomarkers for serous ovarian cancer patients (Figure 3a). We additionally employed the GEPIA database to confirm the expression of these candidates in serous ovarian cancer tissues and nonovarian tumor tissues, indicating that the over-expression of genes was positive relative to serous ovarian cancer (*p* < 0.05). The differential expression of five key genes in ovarian tumors and non-ovarian tumor tissues through the GEPIA dataset are shown in Figure 3b,c. The differential expression in serous ovarian cancer tissues and non-ovarian cancer tissues may also explain the trend we observed from the GEO database. These results therefore clearly demonstrated that the expression of ITGB2, VEGFA, CLDN4, OCLN, and SPP1 were significantly correlated with poorer clinical outcomes in serous ovarian cancer.

### 3.5. ITGB2 Was Associated with TAM (Tumor-Associated Macrophage) Infiltration in Serous Ovarian Cancer

Previous studies [12,13,14] reported that the survival rate could be independently evaluated by the frequency of various tumor-infiltrating lymphocytes in patients bearing a solid malignant tumor as well as a hematologic tumor. We explored the underlying connection between the five candidate genes and their characteristic immune infiltration in a tumor-immune microenvironment via the TIMER and EPIC databases. First, we used the EPIC database to draw up the major immune cell types in two profiles. Interestingly, the results showed that the immune microenvironment appeared to be characterized by a dilemma between immune activation and immune suppression (Appendix A). We next assessed the particular association between five genes and then a variety of infiltrating immune cells in serous ovarian cancer, such as CD4+/CD8+ T cells, B cells, and macrophages. Importantly, we found that ITGB2 in the serous ovarian cancer microenvironment was positively correlated with immune infiltration compared to the other four prognosis-related genes, and it had a strong correlation with the infiltration of macrophages, such as macrophages (r = 0.3107, *p* = 3.17 × 10^−7^), M0 macrophages (r = 0.157, *p* = 1.32 × 10^−2^), M1 macrophages (r = 0.3053, *p* = 1.02 × 10^−8^), and M2 macrophages (r = 0.707, *p* = 4.71 × 10^−39^), in serous ovarian cancer, while it had a moderate correlation with CD4+/CD8+ T cells (cluster of differentiation 4 + Tregs/cluster of differentiation 8 + Tregs) (Appendix A). This feature explains why immune activation and suppression coexist in the ovarian cancer microenvironment (Figure 4a,b). In addition, we also probed into the relationship between the five key genes and a variety of the immune cell-surface genes of TAMs, M0 macrophages, M1 macrophages, and M2 macrophages via the GEPIA database, which indicated a positive correlation between ITGB2 expression and most of the genetic surface markers in macrophages, M1 macrophages, M2 macrophages, and TAMs (Appendix A). However, there was no significant correlation between the other four genes with macrophage infiltration. For this reason, we selected ITGB2 as the top candidate prognostic biomarker and focused on the role of ITGB2 in the serous ovarian cancer microenvironment. Additionally, ITGB2 has the potential to be an indicator of pan-cancer immune infiltration (Figure 4c). The above findings in our study implicate that ITGB2 may affect the prognosis of serous ovarian cancer patients, possibly through remodeling the tumor immune microenvironment.

### 3.6. Validation the Expression of ITGB2 in Serous Ovarian Cancer

Having demonstrated the potential relationship between ITGB2 and the immune infiltration in serous ovarian cancer patients via an integrated bioinformatics analysis, we deduced that this association may also be applicable to serous ovarian cancer patients. Hence, for the sake of verifying the possible correlation between ITGB2 and serous ovarian cancer, we evaluated the expression and cellular localization of ITGB2 in serous ovarian cancer and non-ovarian tumor cells or tissues via a series of experiments. Consistent with the previous findings, the expression of ITGB2 was increased significantly in serous ovarian cancer cells compared to normal ovarian epithelial cells both in protein and mRNA (messenger ribonucleic acid) levels. Results show that the relative expression of ITGB2 mRNA in the patient-derived ovarian cancer cell (PDC3) was 1.625, which was significantly upregulated (*p* = 0.0304 < 0.05) compared with the expression in the IOSE80 cell (mRNA = 1). (Figure 5b,c). Additionally, ITGB2 was found to be positively expressed (27.27%) in the cytoplasm in serous ovarian cancer cells and tissues (Figure 5a–d) but was almost negative (5.93%) in the normal ovarian epithelial cells and non-tumor tissues, with a significant difference (*p* < 0.0001) (Figure 5e). This indicates that the upregulation of ITGB2 might play a role in serous ovarian cancer.

### 3.7. Immune-Associated Pathways for ITGB2

Since the upregulated ITGB2 was connected with the immune infiltration in serous ovarian cancer, we further applied Wikipathways and Reactome analyses for the potential molecular mechanism of ITGB2. The Reactome analysis revealed that ITGB2 was most related to an adaptive and innate immune system in the host. ITGB2 could influence the immunoregulatory crosstalk between a lymphoid and a non-lymphoid cell in the adaptive immune system in host while participating in FCGR (Fc-gamma receptor)-dependent phagocytosis in the innate immune system. The common pathways between the innate and adaptive immune system, including integrin cell-surface interactions, toll-like receptor cascades, neutrophil degranulation, interleukin-4, and interleukin-13 signaling. In addition to, the WikiPathways analysis also showed that the PI3K/Akt/mTOR signaling pathway (phosphatidylinositol 3 kinase-protein kinase b-mammalian target of rapamycin) was the most involved in focal adhesion and the integrin-mediated cell adhesion pathways (Appendix A). Cause cell adhesion and focal adhesion are associated with the immune system; these results encouraged us to further probe the regulatory effects of ITGB2 on the immune system, and the mechanism of the function might have relevance to the influence of the PI3K/Akt/mTOR signaling pathway.

## 4. Discussion

Despite tremendous progress in immunotherapy, epithelial ovarian cancer patients remain poorly responsive to it, probably due to immunosuppression and the high heterogeneity. Therefore, to enhance the clinical efficacy of immunotherapy by heating the “cold” ovarian cancer, further studies on new therapies and the molecular mechanisms in the ovarian cancer tumor immune microenvironment remain to be elucidated [15,16].

The tumor microenvironment consists of abundant stromal components, which are composed of various cell types, including the extracellular matrix, fibroblasts, chondrocytes, and mesothelial cells, non-stromal components comprising different immune cells and adipocytes, and microbiota, such as mycoplasma [17,18]. Indubitably, the diverse cellular compositions play a crucial role in the heterogeneity of serous ovarian cancers and induce the initiation, progression, and resistance to the anti-tumor therapy of serous ovarian cancer [19,20]. Macrophages are plastic mononuclear phagocytic cells which can polarize into specific functional phenotypes with the stimulation of some cytokines. TAMs, originating from tissue-resident macrophages, are the most predominant subgroup of immune cells in the ovarian cancer microenvironment. TAMs usually play a pro-tumorigenic role in the tumor microenvironment, functioning to drive tumor proliferation and metastasis [21]. This could polarize into tumoricidal M1 macrophages and tumorigenic M2 macrophages. M2 macrophages commonly predominate in the ovarian cancer environment [22,23]. Thus, it may be crucial to reprogram the TAMs in TME to improve the strong immunosuppression and increase therapeutic effectiveness of ovarian cancer, with the aim of providing a new insight for immunotherapy in ovarian cancer [24,25]. Interestingly, based on the assessments of the TIMER and EPIC databases, we found that the tumor environment comprised CAFs (cancer-associated fibroblasts), CD4+/CD8+ T cells, and macrophages, revealing that the immune microenvironment of ovarian cancer presents a dichotomy between immune activation and suppression. This dynamic characteristic explains why the high expression of ITGB2 is accompanied by the immune activation, but does not reverse carcinogenesis [26]. Across the databases, we also found that ITGB2 has a positive relevance to immune infiltration in a various of cancer types, including ovarian cancer. This study consistently observed that ITGB2 was moderately connected with the CD4+/CD8+ T cell and B cell infiltration, while it was significantly associated with macrophage infiltration. An additional key finding was that the ITGB2 expression was significantly higher in M2 macrophages compared with M0 macrophages and M1 macrophages. This implies that ITGB2 might play a decisive role in regulating TAMs differentiation.

ITGB2, also known as CD18/LFA-1, is a transmembrane cell surface receptor attached to the integrin family. It can encode integrin beta chains and bind to the alpha chains, forming the versatile integrin heterodimers. ITGB2 can be involved in cell adhesion as well as the cell-surface mediated signaling pathway, functioning as hemidesmosomes in the transition and maturation in a various of immune cells and contributing to the interplay between the immune system and the organism, thereby motivating tumorigenesis. [27,28,29]. Additionally, studies by Grabbe showed that Tregs adhere to DCs (dendritic cells) via ITGB2, leading to an impaired antigen presentation ability and the inhibition of T-cells, revealing the possible mechanism of T cell activation [30]. ITGB2 plays a vital role in in various disorders covering nasopharyngeal carcinoma, NSCLC (non-small cell lung cancer), glioma, breast cancer, and osteosarcoma, and non-tumor diseases, including the SSc (systemic sclerosis) and NEC (necrotizing enterocolitis) of infants [31]. Research by Wang’s group [32] demonstrated that the ITGB2/FAK/SOX6 (focal adhesion kinase/SRY-box containing gene 6) pathway was activated to promote metastasis, invasion, and glycolysis in nasopharyngeal carcinoma via phosphorylation and protein interactions, which may offer a novel target for the therapy of nasopharyngeal carcinoma. Additionally, previous research also reported that ITGB2 was upregulated in osteosarcoma, [33] which could drive tumor metastasis via the ITGB2/FAK pathway [34]. Xu [35] reported that the expression of ITGB2 in NSCLC was associated with Treg cells and MDSC (myeloid-derived suppressor cell) infiltration positively. They found that the upregulation of ITGB2 in NSCLC cell lines increased the expression of immune-related proteins, such as N-cadherin, snail, and slug, while decreasing the E-cadherin expression, laying the foundation for further research on immunotherapy. Regarding the non-tumor disorders, studies suggested that the upregulated mRNA expression level of ITGB2 in PBMCs (peripheral blood mononuclear cells) did not associate with disease severity of SSc patients, nor did the status in premature infants with NEC [36,37]. Our results highlight the potential ability of ITGB2 to be a regulator of immune infiltration and a marker of the response to immunotherapy for ovarian cancer patients.

## 5. Conclusions

Generally, we conducted an integrated bioinformatic analysis of genome expression and immune characteristics in serous ovarian cancer with validation in different experiments. We revealed that the expression of ITGB2, VEGFA, CLDN4, OCLN, and SPP1 were increased in serous ovarian cancer tissues compared with non-ovarian tumor ovarian tissues, and that the upregulation of these genes was associated with immune infiltration and a poor clinical outcome in serous ovarian cancer patients. Additionally, ITGB2 might function as a novel prognostic biomarker for immunotherapy and might work as a biomarker for efficacy prediction, monitoring toxic and adverse effects of immunotherapy, and even to screen the refractory patients suitable for immunotherapy. Furthermore, we identified the specific expression of ITGB2 in both ovarian cancer patient-derived cells and tumor tissues. Thus, this study identified, for the first time, that ITGB2 may act as a novel prognostic immunomarker for ovarian cancer patients.

## Figures and Tables

**Figure 1 diagnostics-13-01169-f001:**
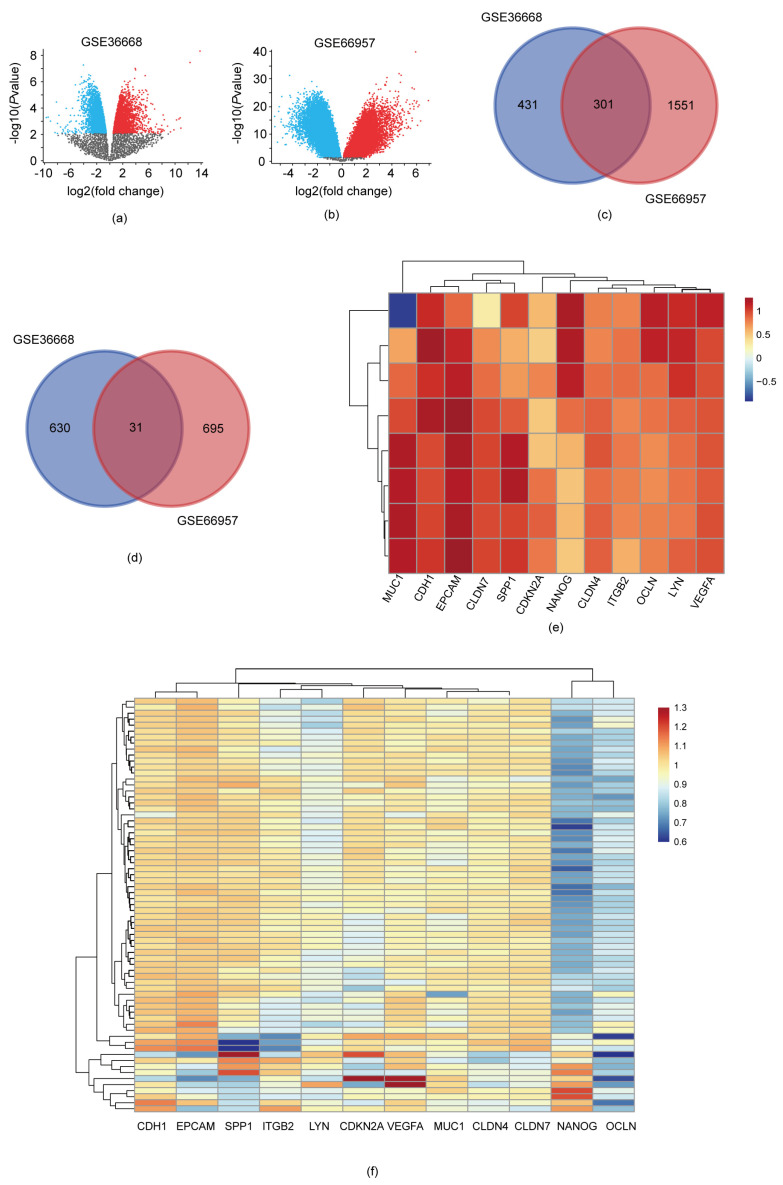
DEGs of two expression profiles. (**a**) Differentially expressed genes in profile GSE36668, identified with volcano plot; (**b**) Differentially expressed genes in profile GSE66957, identified with volcano plot. (The blue dots indicate downregulated genes in serous ovarian cancer patients, and red dots indicate upregulated genes in serous ovarian cancer patients); (**c**) Venn analysis of the up-regulated genes in two profiles; (**d**) Venn analysis of the down-regulated genes in two profiles; (**e**) Heatmap representing several selected DEGs between ovarian cancer and control groups in GSE36668 profile (one sample per raw); (**f**) Heatmap representing several selected DEGs between ovarian cancer and control group in GSE66957 profile (one sample per raw).

**Figure 2 diagnostics-13-01169-f002:**
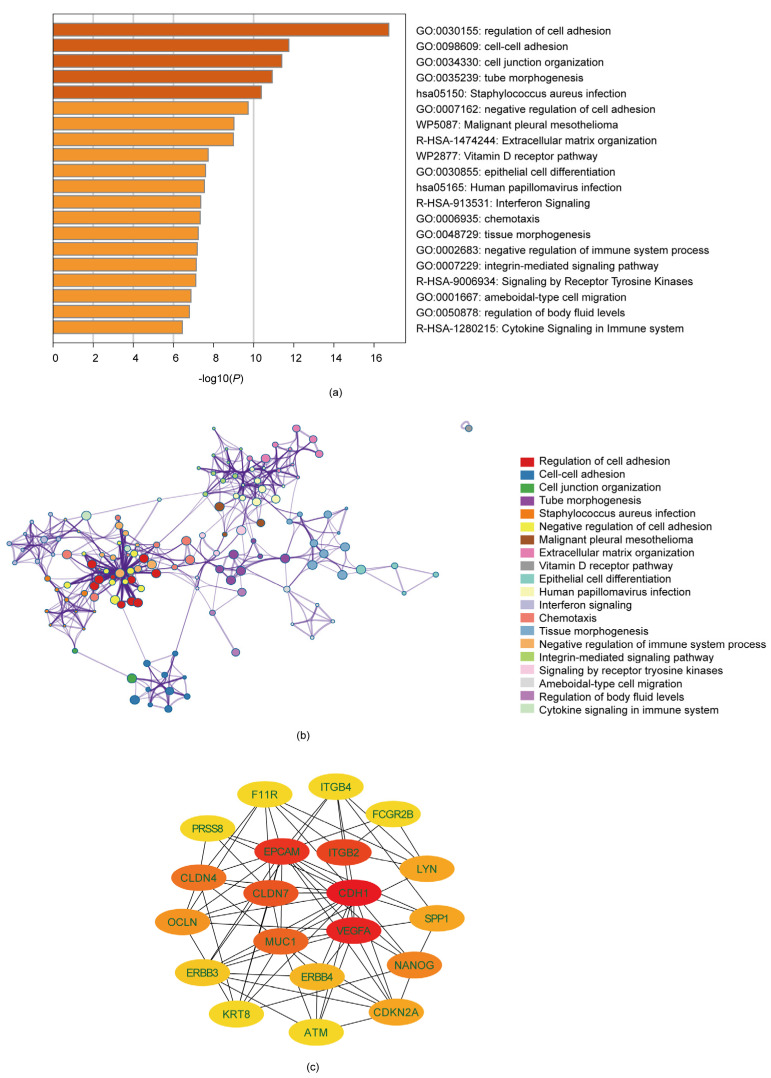
Enrichment analysis of the DEGs. (**a**) Top 20 functional enrichment results from the enrichment analyses of DEGs screened by Metascape; (**b**) The visualization of top 20 enriched terms of DEGs; each specific color is indicated with a cluster ID; (**c**) Hub genes’ interaction network constructed by Cytoscape. The score is exhibited in the color orange: a darker color usually indicates a higher score in this network.

**Figure 3 diagnostics-13-01169-f003:**
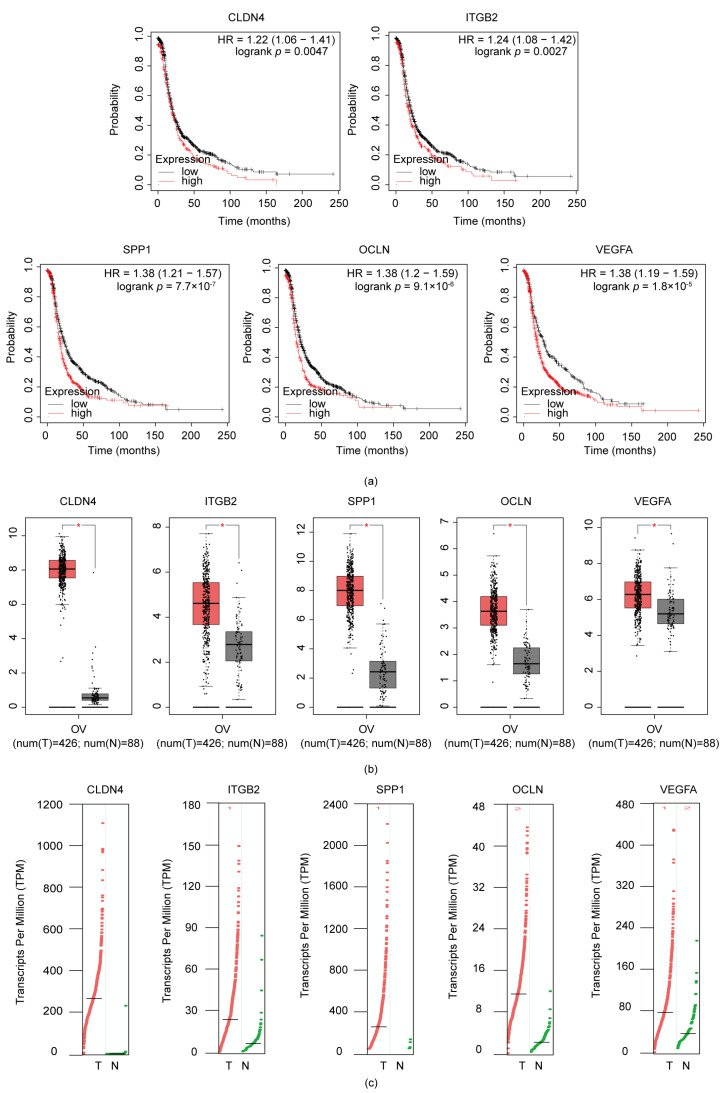
Potential expression of prognostic hub genes in ovarian cancer. (**a**) ITGB2, VEGFA, CLDN4, OCLN, and SPP1 were correlated with unfavorable progression free survival for serous ovarian cancer patients; (**b**,**c**) Expression of ITGB2, VEGFA, CLDN4, OCLN, and SPP1 in serous ovarian cancers, compared with non-ovarian tumor tissues (* *p* < 0.05).

**Figure 4 diagnostics-13-01169-f004:**
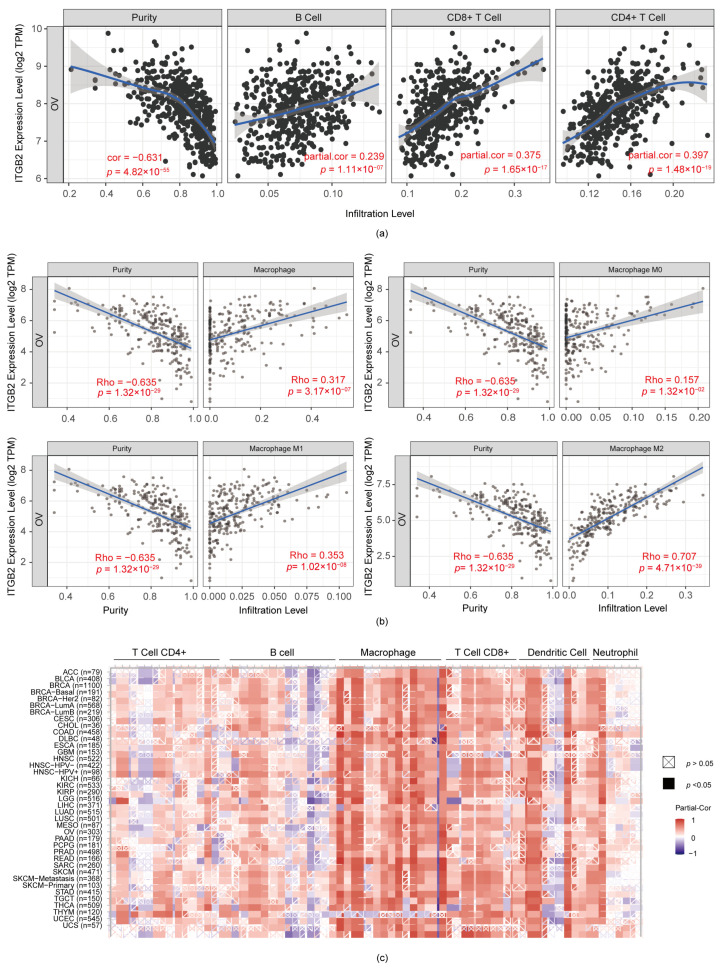
Relevance between ITGB2 expression and infiltration of classical immune cells. (**a**) Relevance between ITGB2 and a variety of infiltrating immune cells (B cells, CD4+/CD8+ T cells); (**b**) Relevance between ITGB2 and macrophages, including: macrophages, M0 macrophages, M1 macrophages, and M2 macrophages; (**c**) Relevance between ITGB2 expression and classical immune cells in other cancer types in patients.

**Figure 5 diagnostics-13-01169-f005:**
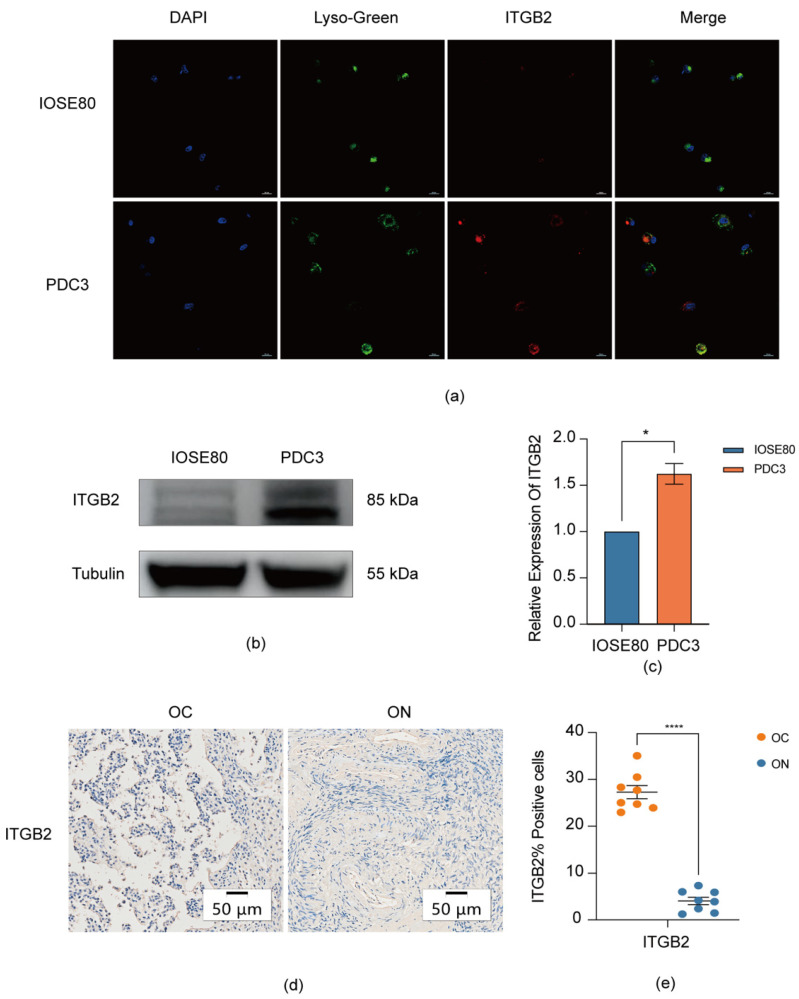
Validation of the expression of ITGB2 in PDC and specimens. (**a**) Location of ITGB2 in two ovarian cancer patient-derived cells via confocal microscopy (red: ITGB2; blue: cell nucleus; green: cell lysosome; scale bar = 20 µm); (**b**) Expression of ITGB2 in IOSE80 and PDC3 cells by Western blotting; (**c**) Detection the relative mRNA expression of ITGB2 via qPCR (* *p* < 0.05); (**d**) Representative immunohistochemical staining of ITGB2 in serous ovarian cancer tissues and control tissues. (OC: ovarian cancer; ON: matched control ovarian tissues); (**e**) Quantification of ITGB2-positive cells in tumor tissues (**** *p* < 0.0001, two-tailed, unpaired Student’s *t* test). At least two independent experiments were performed, and the data in the figure are shown as means ± SEM.

## Data Availability

The bioinformatics data we used were obtained from online database including GEO, GEPIA, Kaplan–Meier plotter, EPIC, and TIMER2.0.

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
