# Peer review of "Identifying ITGB2 as a Potential Prognostic Biomarker in Ovarian Cancer"

_diagnostics, 2023, doi:10.3390/diagnostics13061169_

Round 1

Reviewer 1 Report

In the manuscript entitled "Identifying ITGB2 as Potential Prognostic Biomarkers in Ovarian Cancer" the authors investigated the bioinformatic analysis of ITGB2 and the immune characteristics in ovarian tumorigenesis. Their results revealed that ITGB2 could act as a prognostic biomarker in ovarian cancer. Several important issues should be taken into consideration.

1. Statistical analysis was missing in some experiments. The authors do not state the histograms' statistical significance.

2. Figure 5 needs to be replaced by resonant figures.

3. The findings-based informative conclusion is lacking. The authors would clearly indicate the contribution of their findings to the field.

4. The abbreviations should be further checked and provide the full title at the first appearance, and the writing and grammar need to improve.

5. In the manuscript, the following references may be considered: DOI: 10.1007/s11307-022-01795-1 DOI: 10.1007/s10565-021-09600-5

Author Response

Dear reviewer,

We feel great thanks for your professional review work on our article. As you are concerned, there are several problems that need to be addressed. According to your nice suggestions, we have made extensive corrections to our previous draft, the detailed corrections are listed below. And the revised version has been attached to the reply.

1.Statistical analysis was missing in some experiments. The authors do not state the histograms' statistical significance.

We have supplemented the histograms' statistical significance in our manuscript according to your suggestion. ( 3.6. Validation the expression of ITGB2 in serous ovarian cancer. Line276-292).

  1. Figure 5 needs to be replaced by resonant figures.

I am so sorry that we don’t understand the meaning of “resonant figures”, but we have rechecked the figure5 and replace the figure5 with original vector-graph.

  1. The findings-based informative conclusion is lacking. The authors would clearly indicate the contribution of their findings to the field.

We have re-written this part according to your suggestion. (5. Conclusions, line403-416).

  1. The abbreviations should be further checked and provide the full title at the first appearance, and the writing and grammar need to improve.

We have checked our manuscript and re-written the full title at the first appearance, and have improved the writing and grammar of our manuscript.

  1. In the manuscript, the following references may be considered: DOI: 10.1007/s11307-022-01795-1 DOI: 10.1007/s10565-021-09600-5

We have added the references in our manuscript. (Ref 18 and Ref 32).

Yours

Liu

Reviewer 2 Report

I appreciate the interest in reviewing this publication, it is very interesting and very well managed. However, the author already has very similar papers on other types of cancer, so interest is lost a bit. Throughout the document, the author cites himself several times, so I suggest supporting his findings with publications by other authors in the field. Although he also confirms the role of ITGB2 in the diagnosis of ovarian cancer and its application in other types of cancer.

Author Response

Dear reviewer,

We feel great thanks for your professional review work on our article. As you are concerned, there are several problems that need to be addressed. According to your nice suggestions, we have made extensive corrections to our previous draft, the detailed corrections are listed below. And the revised version has been attached to the reply.

  1. We have added some references in the introduction and discussion to introduce the background in detail. (Line34-53, Ref 3-4,16-18).
  2. We also added some references by other authors in the discussion to support our findings. (Line331-401,Ref 24-26,30-32,35-36).

Yours

Liu

Round 2

Reviewer 1 Report

The Authors have addressed all of my concerns with the original manuscript.